# CROSS-ATTENTIONAL AUDIO-VISUAL FUSION FOR WEAKLY-SUPERVISED ACTION LOCALIZATION

**Juntae Lee, Mihir Jain, Hyoungwoo Park & Sungrack Yun**
Qualcomm AI Research.*
{juntlee,mijain,hwoopark,sungrack}@qti.qualcomm.com

## ABSTRACT

Temporally localizing actions in videos is one of the key components for video understanding. Learning from weakly-labeled data is seen as a potential solution towards avoiding expensive frame-level annotations. Different from other works which only depend on visual-modality, we propose to learn richer audio-visual representation for weakly-supervised action localization. First, we propose a multi-stage cross-attention mechanism to collaboratively fuse audio and visual features, which preserves the intra-modal characteristics. Second, to model both foreground and background frames, we construct an open-max classifier which treats the background class as an open-set. Third, for precise action localization, we design consistency losses to enforce temporal continuity for the action-class prediction, and also help with foreground-prediction reliability. Extensive experiments on two publicly available video-datasets (AVE and ActivityNet1.2) show that the proposed method effectively fuses audio and visual modalities, and achieves the state-of-the-art results for weakly-supervised action localization.

## 1 INTRODUCTION

The goal of this paper is to temporally localize actions and events of interest in videos with weak-supervision. In the weakly-supervised setting, only video-level labels are available during the training phase to avoid expensive and time-consuming frame-level annotation. This task is of great importance for video analytics and understanding. Several weakly-supervised methods have been developed for it (Nguyen et al., 2018; Paul et al., 2018; Narayan et al., 2019; Shi et al., 2020; Jain et al., 2020) and considerable progress has been made. However, only visual information is exploited for this task and audio modality has been mostly overlooked. Both, audio and visual data often depict actions from different viewpoints (Guo et al., 2019). Therefore, we propose to explore the joint audio-visual representation to improve the temporal action localization in videos.

A few existing works (Tian et al., 2018; Lin et al., 2019; Xuan et al., 2020) have attempted to fuse audio and visual modalities to localize *audio-visual events*. These methods have shown promising results, however, these audio-visual events are essentially actions that have strong audio cues, such as playing guitar, and dog barking. Whereas, we aim to localize wider range of actions related to sports, exercises, eating etc. Such actions can also have weak audio aspect and/or can be devoid of informative audio (e.g. with unrelated background music). Therefore, it is a key challenge to fuse audio and visual data in a way that leverages the mutually complementary nature while maintaining the modality-specific information.

In order to address this challenge, we propose a novel multi-stage cross-attention mechanism. It progressively learns features from each modality over multiple stages. The inter-modal interaction is allowed at each stage only through cross-attention, and only at the last stage are the visually-aware audio features and audio-aware visual features concatenated. Thus, an audio-visual feature representation is obtained for each snippet in videos.

Separating background from actions/events is a common problem in temporal localization. To this end, we also propose: (a) foreground reliability estimation and classification via open-max classifier and (b) temporal continuity losses. First, for each video snippet, an open-max classifier predicts

---

*Qualcomm AI Research is an initiative of Qualcomm Technologies, Inc.

scores for action and background classes, which is composed of two parallel branches for action classification and foreground reliability estimation. Second, for precise action localization with weak supervision, we design temporal consistency losses to enforce temporal continuity of action-class prediction and foreground reliability.

We demonstrate the effectiveness of the proposed method for weakly-supervised localization of both audio-visual events and actions. Extensive experiments are conducted on two video datasets for localizing audio-visual events (AVE[1]) and actions (ActivityNet1.2[2]). To the best of our knowledge, it is the first attempt to exploit audio-visual fusion for temporal localization of unconstrained actions in long videos.

## 2 RELATED WORK

Our work relates to the tasks of localizing of actions and events in videos, as well as to the regime of multi-model representation learning.

**Weakly-supervised action localization:** Wang et al. (2017) and Nguyen et al. (2018) employed multiple instance learning (Dietterich et al., 1997) along with attention mechanism to localize actions in videos. Paul et al. (2018) introduced a co-activity similarity loss that looks for similar temporal regions in a pair of videos containing a common action class. Narayan et al. (2019) proposed center loss for the discriminability of action categories at the global-level and counting loss for separability of instances at the local-level. To alleviate the confusion due to background (non-action) segments, Nguyen et al. (2019) developed the top-down class-guided attention to model background, and (Yu et al., 2019) exploited temporal relations among video segments. Jain et al. (2020) segmented a video into interpretable fragments, called ActionBytes, and used them effectively for action proposals. To distinguish action and context (near-action) snippets, Shi et al. (2020) designed the class-agnostic frame-wise probability conditioned on the attention using conditional variational auto-encoder. Luo et al. (2020) proposed an expectation-maximization multi-instance learning framework where the key instance is modeled as a hidden variable. All these works have explored various ways to temporally differentiate action instances from the near-action background by exploiting only visual modality, whereas we additionally utilize audio modality for the same objective.

**Audio-visual event localization:** The task of audio-visual event localization, as defined in the literature, is to classify each time-step into one of the event classes or background. This is different from action localization, where the goal is to determine the start and the end of each instance of the given action class. In (Tian et al., 2018), a network with audio-guided attention was proposed, which showed prototypical results for audio-visual event localization, and cross-modality synchronized event localization. To utilize both global and local cues in event localization, Lin et al. (2019) conducted audio-visual fusion in both of video-level and snippet-level using multiple LSTMs. Assuming single event videos, Wu et al. (2019) detected the event-related snippet by matching the video-level feature of one modality with the snippet-level feature sequence of the other modality. Contrastingly, our cross-attention is over the temporal sequences from both the modalities and does not assume single-action videos. In order to address the temporal inconsistency between audio and visual modalities, Xuan et al. (2020) devised the modality sentinel, which filters out the event-unrelated modalities. Encouraging results have been reported, however, the localization capability of these methods has been shown only for the short fixed-length videos with distinct audio cues. Differently, we aim to fuse audio and visual modalities in order to also localize actions in long, untrimmed and unconstrained videos.

**Deep multi-modal representation learning:** Multi-modal representation learning methods aim to obtain powerful representation ability from multiple modalities (Guo et al., 2019). With the advancement of deep-learning, many deep multi-modal representation learning approaches have been developed. Several methods fused features from different modalities in a joint subspace by outer-product (Zadeh et al., 2017), bilinear pooling (Fukui et al., 2016), and statistical regularization (Aytar et al., 2017). The encoder-decoder framework has also been exploited for multi-modal learning for image-to-image translation (Huang et al., 2018) and to produce musical translations (Mor et al.,

---

[1]https://github.com/YapengTian/AVE-ECCV18
[2]http://activity-net.org/download.html

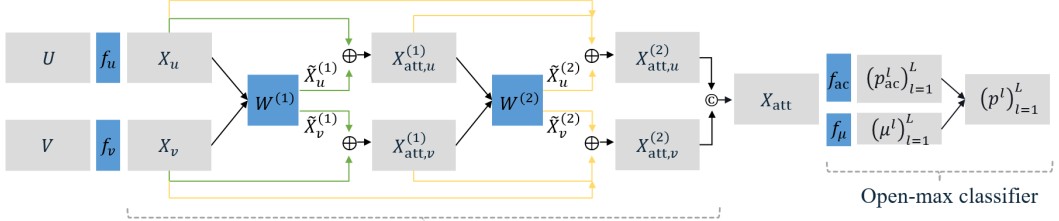

Multi-stage cross-attention

Figure 1: **The proposed architecture** has two parts: modality fusion and open-max classification. (a) *Fusion by multi-stage cross-attention*: The input audio ($U$) and visual ($V$) features are embedded by the two fully-connected layers $f_u$ and $f_v$, and passed through the multiple stages of the cross-attention. At the $t$th stage, the attended audio-visual embeddings, $X^{(t)}_{\mathrm{att},u}$ and $X^{(t)}_{\mathrm{att},v}$, are calculated using the results from the previous stages through dense skip connections, and activated by a non-linear function. Here, ⓒ and ⊕ denote concatenation and summation operations. This figure shows 2-stage case. The dense skip connections of two stages are depicted as green and yellow arrows, respectively. At the last stage, two attended features are concatenated. (b) *Open-max classifier* takes the concatenated audio-visual features as input and generates classification scores for action classes and background. More detailed description is given in Appendix C.

2018). Another category of approaches aim to disjointly learn the features of each modality under cross-modality constraints such as cross-modal ranking (Frome et al., 2013; Lazaridou et al., 2015; Kiros et al., 2014) or feature distance (Pan et al., 2016; Xu et al., 2015; Liong et al., 2016). Our approach belongs to this category and uses cross-correlation as cross-modality constraint. Cross-correlation has been exploited to generate visual features attended by text for visual question answering (Kim et al., 2017; Yu et al., 2017). It has also been used to obtain cross-attention for few-shot learning (Hou et al., 2019) and image-text matching (Lee et al., 2018; Wei et al., 2020). In our work, we adopt the cross-correlation to generate both of audio and visual features attended by each other. The most similar to our cross-attention mechanism is the cross-attention module of Hou et al. (2019), which computes cross-correlation spatially between features maps of two images (sample and query). Whereas, our cross-attention is designed for video and is computed between two temporal sequences of different modalities.

## 3    METHODOLOGY

In this section, we introduce the proposed framework for weakly-supervised action and event localization. Fig. 1 illustrates the complete framework. We first present the multi-stage cross-attention mechanism to generate the audio-visual features in Sec. 3.1. Then, we explain open-max classification to robustly distinguish the actions[3] from unknown background in 3.2. Finally, in Sec. 3.3, we describe the training loss including two consistency losses designed to enforce temporal continuity of the actions and background.

**Problem statement:** We suppose that a set of videos only with the corresponding video-level labels are given for training. For each video, we uniformly sample $L$ non-overlapping snippets, and then extract the audio features $U = (u^l)^L_{l=1} \in \mathbb{R}^{d_u \times L}$ with a pre-trained network, where $u^l$ is the $d_u$ dimensional audio feature representation of the snippet $l$. Similarly, the snippet-wise visual features $V = (v^l)^L_{l=1} \in \mathbb{R}^{d_v \times L}$ are also extracted. The video-level label is represented as $c \in \{0, 1, \dots, C\}$, where $C$ is the number of action classes and 0 denotes the background class. Starting from the audio and visual features, our approach learns to categorize each snippet into $C + 1$ classes and hence localizes actions in weakly-supervised manner.

### 3.1    MULTI-STAGE CROSS-ATTENTION MECHANISM

While multiple modalities can provide more information than a single one, the modality-specific information may be reduced while fusing different modalities. To reliably fuse the two modalities,

---

[3]For brevity we refer both action and event as action.

we develop the *multi-stage cross-attention* mechanism where features are separately learned for each modality under constraints from the other modality. In this way, the learned features for each modality encodes the inter-modal information, while preserving the exclusive and meaningful intra-modal characteristics.

As illustrated in Fig. 1, we first encode the input features $U$ and $V$ to $X_u = (x_u^l)_{l=1}^L$ and $X_v = (x_v^l)_{l=1}^L$ via the modality-specific fully-connected (FC) layers $f_u$ and $f_v$, where $x_u^l$ and $x_v^l$ are in $\mathbb{R}^{d_x}$. After that, we compute the cross-correlation of $X_u$ and $X_v$ to measure inter-modal relevance. To reduce the gap of the heterogeneity between the two modalities, we use a learnable weight matrix $W \in \mathbb{R}^{d_x \times d_x}$ and compute the cross-correlation as

$$\Lambda = X_u^T W X_v \tag{1}$$

where $\Lambda \in \mathbb{R}^{L \times L}$. Note that $x_u^l$ and $x_v^l$ are $l_2$-normalized before computing the cross-correlation.

In the cross-correlation matrix, a high correlation coefficient means that the corresponding audio and visual snippet features are highly relevant. Accordingly, the $l$th column of $\Lambda$ corresponds to the relevance of $x_v^l$ to $L$ audio snippet features. Based on this, we generate the cross-attention weights $A_u$ and $A_v$ by the column-wise soft-max of $\Lambda$ and $\Lambda^T$, respectively. Then, for each modality, the attention weights are used to re-weight the snippet features to make them more discriminative given the other modality. Formally, the attention-weighted features $\tilde{X}_u$ and $\tilde{X}_v$ are obtained by

$$\tilde{X}_u = X_u A_u \quad \text{and} \quad \tilde{X}_v = X_v A_v. \tag{2}$$

Note that each modality guides the other one through the attention weights. This is to ensure the meaningful intra-modal information is well-preserved while applying the cross-attention.

To extensively delve into cross-modal information, we repeatedly apply the cross-attention multiple times. However, during the multi-stage cross-attention, the original modality-specific characteristics may be over-suppressed. To prevent this, we adopt the dense skip connection (Huang et al., 2017). More specifically, at stage $t$, we obtain the attended audio features by

$$X_{\text{att},u}^{(t)} = \tanh\left(\sum_{i=0}^{t-1} X_{\text{att},u}^{(i)} + \tilde{X}_u^{(t)}\right) \tag{3}$$

where $X_{\text{att},u}^{(0)}$ is $X_u$, and $\tanh(\cdot)$ denotes the hyperbolic tangent activation function. Similar to $X_{\text{att},u}^{(t)}$, the attended visual features $X_{\text{att},v}^{(t)}$ are generated for the visual modality.

At the last stage $t_{\text{e}}$, we concatenate the attended audio and visual features to yield audio-visual features,

$$X_{\text{att}} = [X_{\text{att},u}^{(t_{\text{e}})}; X_{\text{att},v}^{(t_{\text{e}})}] \tag{4}$$

where $t_{\text{e}}$ is empirically set to 2 which will be discussed in the ablation studies in Section 4.3.

**Discussion** Applying the cross-attention (Eq. 2) brings the audio and visual embeddings closer, while the skip connections (Eq. 3) enforce modality specific information, more so with dense skip connections. Using both the cross-attention and the dense skip connections alternatively over multiple stages, we progressively learn optimal embeddings for fusion. Learning in this way, we aim to achieve right amount of compatibility between the two embeddings while preserving the modality specific information, in order to optimize for the training objective.

## 3.2 OPEN-MAX CLASSIFICATION

Video segments can be dichotomized into foreground actions and background. For precise action localization, distinguishing the background from the actions is important as well as categorizing the action classes. However, unlike action classes, the background class comprises of extremely diverse types of non-actions. Therefore, it is not possible to train for the wide range of background classes that the model may confront at the test time.

To resolve this problem, we address the background as an open set (Dietterich, 2017; Bendale & Boult, 2016). As illustrated in Fig. 1, we construct an open-max classifier on top of the multi-stage cross-attentional feature fusion. Specifically, the open-max classifier consists of two parallel

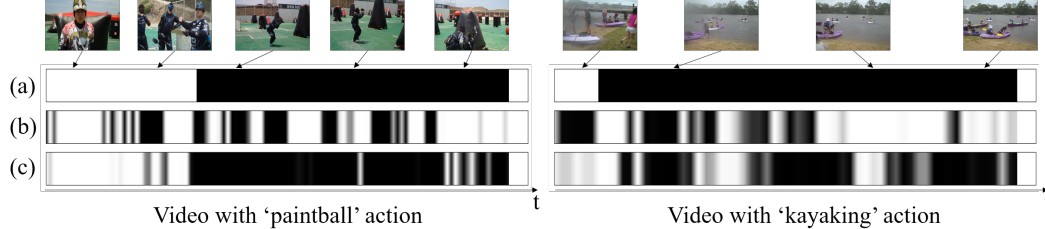

Figure 2: Visualization of class activation sequences for the target actions in two example videos: The ground-truth segments are shown in (a). The class activation sequences obtained without $\mathcal{L}_{\text{cont}}$ and $\mathcal{L}_{\text{pseu}}$ are shown in (b), which improve and get better aligned to the ground-truth segments when these continuity losses are used as shown in (c). The activation is depicted in gray-scale, where lower intensity indicates more strong activation.

**FC layers for action classification and foreground reliability estimation.** The attended audio-visual feature $x_{\text{att}}^l$, where $l = 1, \ldots, L$, is fed snippet-wise into the open-max classifier. The first FC layer outputs a snippet-wise activation vector $h^l = [h^l(1), \ldots, h^l(C)]$ for $C$ action classes, which is converted to probability scores, $p_{\text{ac}}^l$ by soft-max function.

Simultaneously, the second FC layer is applied on $x_{\text{att}}^l$, followed by a sigmoid function to estimate its *foreground reliability*, $\mu^l \in [0, 1]$. The foreground reliability, $\mu^l$, is the probability of snippet $l$ belonging to any action class. The low reliability indicates that no action occurs in the snippet. Therefore, we compute the probability for the background class as the complement of $\mu^l$, by $p_{\text{bg}}^l = 1 - \mu^l$.

Lastly, the open-max classifier outputs the probability distribution $p^l$ over $C + 1$ classes including the background and $C$ actions as

$$p^l = [p_{\text{bg}}^l;\; \mu^l p_{\text{ac}}^l]. \tag{5}$$

## 3.3 TRAINING LOSS

Next, we describe the loss functions to train our model. The actions or foreground do not abruptly change over time. To impose this constraint, we devise two types of *temporal continuity* losses.

**Foreground continuity loss:** Foreground continuity implies two important properties for neighboring snippets: (a) similar foreground reliability in a class-agnostic way, and (b) consistent open-max probabilities for a target foreground class.

The first of the two constraints is imposed via class-agnostic foreground continuity:

$$\mu_{\text{ag}}^l = \frac{1}{B+1} \sum_{i=-B/2}^{B/2} G(i)\, \mu^{l-i} \tag{6}$$

where $G(i)$ is a Gaussian window of width $B + 1$ to apply temporal smoothing over foreground reliability around $l$th snippet. For the second constraint, temporal Gaussian smoothing is applied over open-max probability of video-level ground-truth action class, $\hat{c}$, to obtain class-specific foreground continuity:

$$\mu_{\text{sp}}^l = \frac{1}{B+1} \sum_{i=-B/2}^{B/2} G(i)\, p^{l-i}(\hat{c}) \tag{7}$$

Finally, the foreground continuity loss is defined as:

$$\mathcal{L}_{\text{cont}} = \frac{1}{L} \sum_{l=1}^{L} |\mu^l - \mu_{\text{ag}}^l| + |\mu^l - \mu_{\text{sp}}^l|. \tag{8}$$

The foreground continuity loss imposes temporal continuity of foreground, and hence also helps in separating background from the action classes.

Table 1: **Ablation for multi-stage cross-attention.** The results for different stages of the cross-attention are reported for the AVE and ActivityNet1.2 datasets. The comparison with the uni-modal approach shows the impact of leveraging the multi-modality and the cross-attention.

|  |  |  | Uni-modal | | Multi-modal | | | |
| --- | --- | --- | --- | --- | --- | --- | --- | --- |
|  |  |  | Audio | Visual | 0-stage | 1-stage | 2-stage | 3-stage |
| AVE | Accuracy (%) |  | 32.1 | 45.2 | 65.0 | 75.0 | 77.1 | 75.6 |
|  |  | 0.5 | 12.3 | 38.3 | 37.6 | 42.1 | 44.8 | 39.5 |
|  |  | 0.6 | 10.9 | 32.9 | 32.4 | 35.3 | 37.8 | 33.8 |
| ActivityNet1.2 | mAP@IoU (%) | 0.7 | 9.7 | 25.4 | 26.7 | 29.5 | 30.8 | 27.9 |
|  |  | 0.8 | 7.6 | 19.2 | 19.4 | 20.8 | 22.5 | 20.9 |
|  |  | Avg. | 7.8 | 22.1 | 22.0 | 24.1 | 26.0 | 23.3 |

**Pseudo localization loss:** Here, we consider the action or background class continuity, which implies that the open-max probabilities, $p^l$, agrees with the classification of neighbouring snippets. This can be used to obtain the pseudo label for snippet $l$. We first average the open-max prediction of $N$ neighbor snippets and itself, $q^l = \frac{1}{N+1} \sum_{i=l-N/2}^{l+N/2} p^i$. We set $\hat{c}^l = \arg\max_c(q^l(c))$ as the pseudo label, but only retain it when the largest class probability of $q^l$ is higher than a predefined threshold $\tau$. Accordingly, the pseudo localization loss is formulated by

$$\mathcal{L}_{\text{pseu}} = \frac{1}{L} \sum_{l=1}^{L} \mathbb{1}(\max(q^l) \geq \tau)(-\log p^l(\hat{c}^l)) \tag{9}$$

**Total loss:** Additionally, we employ the multiple instance learning (MIL) and co-activity similarity (CAS) losses (Paul et al., 2018). The final loss $\mathcal{L}$ is defined by

$$\mathcal{L} = \mathcal{L}_{\text{mil}} + \alpha\mathcal{L}_{\text{cas}} + \beta\mathcal{L}_{\text{cont}} + \gamma\mathcal{L}_{\text{pseu}} \tag{10}$$

where $\mathcal{L}_{\text{mil}}$ and $\mathcal{L}_{\text{cas}}$ denote MIL and CAS losses, respectively. For details see Appendix D.

Figs. 2 (b) and (c) compare the class activation sequences along the temporal axis for the target classes between the models trained without and with the two consistency losses, respectively. We see that class activations are more continuous in the model with the consistency losses.

## 4 EXPERIMENTS

In this section, we provide experimental analysis and comparative evaluation to show the effectiveness of the proposed method. More experiments and qualitative results are in the Appendix.

### 4.1 DATASETS AND EVALUATION METHOD

**Datasets:** We evaluate our approach on Audio-Visual Event (AVE) and ActivityNet1.2 datasets.

*AVE* dataset is constructed for audio-visual event localization, which contains 3,339 training and 804 testing videos, each lasting 10 seconds with event annotation per second. There are 28 audio-visual event categories covering a wide range of domains, such as animal and human actions, vehicle sounds, and music performance. Each event category has both audio and visual aspects, e.g. church bell, baby crying, man speaking etc.

*ActivityNet1.2* is a temporal action localization dataset with 4,819 train and 2,383 validation videos, which in the literature is used for evaluation. It has 100 action classes of wider variety than AVE dataset, with on an average 1.5 instances per video. The average length of the videos in this dataset is 115 seconds, often with weak audio cues, which makes action localization as well as leveraging audio harder.

**Evaluation metric:** We follow the standard evaluation protocol of each dataset. For the AVE dataset, we report snippet-wise event prediction accuracy. For the ActivityNet1.2 dataset, we generate the action segments (start and end time) from snippet-wise prediction (details are described in the following section), and then measure mean average precision (mAP) at different intersection over union (IoU) thresholds.

Table 2: **Ablations for the consistency losses and open-max classifier.** *Consistency losses:* The lower part of the table shows the impact of each of the two consistency losses, when used with the open-max classifier. *Open-max vs soft-max:* The results for the soft-max are also shown, which demonstrates the advantage of foreground/background modelling by the open-max classification on both the datasets. The model with 2-stage cross-attention is used.

| | Method | $\mathcal{L}_{\text{cont}}$ | $\mathcal{L}_{\text{pseu}}$ | AVE Accuracy (%) | ActivityNet1.2 [mAP@IoU (%)] | | | | |
|---|---|---|---|---|---|---|---|---|---|
| | | | | | 0.5 | 0.6 | 0.7 | 0.8 | Avg. |
| Soft-Max | S-I | | ✓ | 68.5 | 39.4 | 35.7 | 30.7 | 19.8 | 23.8 |
| Open-Max | O-I | ✓ | | 64.9 | 39.9 | 33.7 | 23.8 | 14.3 | 20.3 |
| | O-II | | ✓ | 75.9 | 44.1 | 37.4 | 31.1 | 22.4 | 25.7 |
| | O-III | ✓ | ✓ | 77.1 | 44.8 | 37.8 | 30.8 | 22.5 | 26.0 |

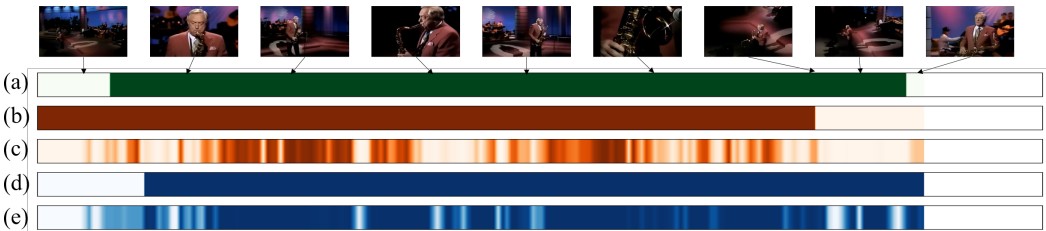

Figure 3: Visualization of the action localization result for an example video from ActivityNet1.2. The ground truth is shown in (a), highlighted in green. The localization and the class activation sequence of the visual-only model are shown in (b) and (c), respectively. Finally, the localization and the class activation sequence for the proposed audio-visual method are shown in (d) and (e).

## 4.2 FEATURE EXTRACTION AND IMPLEMENTATION DETAILS

**Feature extraction:** We use the I3D network (Carreira & Zisserman, 2017) and the ResNet152 architecture (He et al., 2016) to extract the visual features for ActivityNet1.2 and AVE, respectively. The I3D network is pre-trained on Kinetics-400 (Kay et al., 2017), and the features consist of two components: RGB and optical flow. The ResNet 152 is pre-trained on the ImageNet (Russakovsky et al., 2015), and the features are extracted from the last global pooling layer. To extract the audio features, we use the VGG-like network (Hershey et al., 2017), which is pre-trained on the AudioSet (Gemmeke et al., 2017), for both AVE and ActivityNet1.2 datasets.

**Implementation Details:** We set $d_x$ to 1,024, and the LeakyRelu and hyperbolic tangent functions are respectively used for the activation of modality-specific layers and cross-attention modules. In training, the parameters are initialized with Xavier method (Glorot & Bengio, 2010) and updated by Adam optimizer (Kingma & Ba, 2015) with the learning rate of $10^{-4}$ and the batch size of 30. Also, the dropout with a ratio of 0.7 is applied for the final attended audio-visual features. In the loss, the hyper parameters are set as $B = 4$, $\alpha = 0.8$, $\beta = 0.8$ and $\gamma = 1$.

**Localization at test time:** For event localization at test time, i.e. snippet classification, each snippet $l$ is classified into one of event classes (including background) by $\arg\max_c p^l(c)$, where $p^l$ is the open-max probability of snippet $l$. For action localization, we follow the two-stage thresholding scheme of (Paul et al., 2018). The first threshold is applied to filter out the classes that have video-level scores less than the average over all the classes. The second threshold is applied along the temporal axis to obtain the start and the end of each action instance.

## 4.3 ABLATION ANALYSIS

**Multi-stage cross-attention:** To evaluate the effectiveness of the multi-stage cross-attention in audio-visual fusion, we compare two uni-modal methods (audio or visual) and four multi-modal methods with different stages (0-3 stages) on the AVE and ActivityNet1.2 datasets in Table 1. The pseudo-label losses and the open-max classifiers are used in all six cases. In the uni-modal methods, the input feature is embedded using an FC layer, and then fed into the open-max classifier. The 0-stage method denotes a naive fusion, where audio and visual features are fused by simple con-

Table 3: Impact of dense skip connections: Ablation studies on dense skip connection in terms of average of mAP@[0.5:0.05:0.95] for the ActivityNet1.2 dataset. For 2-stage model, no, skip, and dense connections are verified.

| Method | Avg. mAP |
|---|---|
| w/o skip connection | 24.1 |
| w/ skip connection | 24.9 |
| w/ dense skip connection | 26.0 |

Table 4: Comparison of the number of FLOPS and the average mAP@[0.5:0.05:0.95] on the ActivityNet1.2 dataset for visual-only, 1-stage, and 2-stage models. $d_x \times d_x$ are the dimensions for the cross-correlation matrix $W$.

| Method | $d_x$ | No. FLOPS | Avg. mAP |
|---|---|---|---|
| Visual-only | 1024 | $2.3 \times 10^6$ | 22.1 |
| 1-stage | 1024 | $3.5 \times 10^6$ | 24.1 |
| 2-stage | 1024 | $4.0 \times 10^6$ | 26.0 |
| Visual-only | 512 | $1.2 \times 10^6$ | 21.0 |
| 2-stage | 512 | $1.7 \times 10^6$ | 25.9 |

catenation. Even this naive fusion yields higher performance than the uni-modal methods on the AVE dataset. However, that is not the case with more challenging task of the action localization on ActivityNet1.2 dataset. Furthermore, all the later stages improve considerably over 0-stage and the uni-modal cases, for the both datasets. The 2-stage cross-attention achieves the best performance for the both datasets (more in Appendix A). Interestingly, even with the minimal audio cue in ActivityNet1.2 (avg. mAP of audio only is 7.8%), the proposed audio-visual features improve the avg. mAP over visual-only and naive fusion (0-stage) models by 4%.

Fig. 3 shows the qualitative results of the proposed and visual-only models given an example of the ActivityNet1.2 dataset. At the beginning of the video, a performer is shown without any activity. The visual-only model incorrectly predicts the beginning part as a target action while our proposed model correctly predicts it as background. Also, the visual-only model cannot catch the action at the last part of the video since it is visually similar across the frames and has minimal visual activity. Whereas, our model correctly recognizes the last part as an action, owing to the multi-stage cross-attention of effective fusion of the two modalities. More qualitative results are in Appendix E.

**Consistency losses:** We show the ablation over the two proposed losses, $\mathcal{L}_{\text{cont}}$ and $\mathcal{L}_{\text{pseu}}$, while using Open-Max classifier and 2-stage cross-attention, in the lower part of the Table 2. We denote the method with only $\mathcal{L}_{\text{cont}}$ loss by O-I and with only $\mathcal{L}_{\text{pseu}}$ loss by O-II. The proposed method (O-III) with both of the losses performs the best suggesting the importance of both of the losses. Further, O-II outperforms O-I by a big margin on both the datasets, implying that the pseudo localization loss is more critical for the action localization (more in Appendix B.1). This result demonstrates that guiding temporal continuity is essential in the long untrimmed videos as well as the short ones.

**Open-max classifier:** We compare the open-max classifier with the soft-max classifier where the last FC layer outputs activations for $C + 1$ classes are normalized by the soft-max function. As the background is considered a closed set in the soft-max approach, the foreground continuity loss is not available. The soft-max is denoted by S-I in Table 2. Both O-II and O-III versions of the open-max outperform the S-I method with the soft-max. The O-III method improves the accuracy by 8.6% on the AVE dataset and the avg. mAP by 2.2% on the ActivityNet1.2 dataset. For further analysis see Appendix B.2. This shows the advantage of modelling background with the open-max classifier.

**Dense skip connections:** We evaluate the impact of dense skip connections in Table 3 for 2-stage model on the ActivityNet1.2. Compared to no skip connection, performance is improved with the skip connections, and further boosted with the dense skip connection to avg. mAP of 26.0%. This shows preserving the modality specific information leads to better fusion and action localization.

## 4.4 MODEL EFFICIENCY

Though we successfully leverage the audio modality to improve action localization performance, the added modality leads to increased computational cost. The trade-off between efficiency and performance due to the fusion with audio modality is demonstrated in Table 4. When using feature dimension, $d_x = 1024$, the fusion increases the computation over visual-only method by about 52% and 74% after 1-stage and 2-stage, respectively. When we reduce $d_x$ to 512, the visual-only model gets affected while the 2-stage model maintains its performance at 25.9%. Thanks to the effectiveness of the proposed fusion, even with smaller $d_x$ its avg. mAP is well above that of video-only model with $d_x = 1024$, while using about 26% less computation (1.7 MFLOPS vs 2.3 MFLOPS).

Table 5: Comparison of the proposed method with the state-of-the-art fully and weakly-supervised methods (separated by '/') on the AVE dataset. Snippet-level accuracy (%) is reported.

| Method | Tian et al. (2018) | Lin et al. (2019) | Owens & Efros (2018) | Xuan et al. (2020) | Proposed |
|---|---|---|---|---|---|
| Accuracy (%) | 74.7 / 73.1 | 75.4 / 74.2 | 72.3 / 68.8 | 77.1 / 75.7 | - / **77.1** |

Table 6: Comparison of our method with the state-of-the-art action localization methods on the ActivityNet1.2 dataset. The mAPs (%) at different IoU thresholds and average mAP across the IoU thresholds are reported. $\dagger$ indicates audio-visual models. $\star$experiment done using author's code.

| Method | Supervision | mAP@IoU (%) | | | | | | | | | | |
|---|---|---|---|---|---|---|---|---|---|---|---|---|
| | | 0.5 | 0.55 | 0.6 | 0.65 | 0.7 | 0.75 | 0.8 | 0.85 | 0.9 | 0.95 | Avg. |
| Zhao et al. (2017) | Full | 41.3 | 38.8 | 35.9 | 32.9 | 30.4 | 27.0 | 22.2 | 18.2 | 13.2 | 6.1 | 26.6 |
| Paul et al. (2018) | Weak | 37.0 | 33.5 | 30.4 | 25.7 | 16.6 | 12.7 | 10.0 | 7.0 | 4.2 | 1.5 | 18.0 |
| Liu et al. (2019b) | Weak | 37.1 | 33.4 | 29.9 | 26.7 | 23.4 | 20.3 | 17.2 | 13.9 | 9.2 | 5.0 | 21.6 |
| Liu et al. (2019a) | Weak | 36.8 | - | - | - | - | 22.0 | - | - | - | **5.6** | 22.4 |
| Jain et al. (2020) | Weak | 39.4 | - | - | - | 15.4 | - | - | - | - | - | - |
| Shi et al. (2020) | Weak | 41.0 | 37.5 | 33.5 | 30.1 | 26.9 | 23.5 | 19.8 | 15.5 | **10.8** | 5.3 | 24.4 |
| Luo et al. (2020) | Weak | 37.4 | - | - | - | 23.1 | - | - | - | 2.0 | - | 20.3 |
| Tian et al. (2018)$\dagger\star$ | Weak | 15.4 | 13.9 | 12.5 | 11.2 | 10.2 | 9.1 | 7.6 | 5.7 | 1.6 | 0.3 | 8.8 |
| Naive fusion (0-stage)$\dagger$ | Weak | 41.2 | 38.4 | 34.8 | 31.8 | 26.3 | 17.0 | 5.6 | 2.2 | 0.8 | 0.2 | 19.8 |
| Naive fusion (0-stage)$\dagger$ + CL | Weak | 39.4 | 37.0 | 33.5 | 30.6 | 27.7 | 23.6 | 20.0 | 13.6 | 3.0 | 0.6 | 22.9 |
| Ours$\dagger$ | Weak | **44.8** | **42.1** | **37.8** | **34.2** | **30.8** | **26.7** | **22.5** | **15.9** | 4.0 | 1.0 | **26.0** |
| Ours$\dagger$ (efficient) | Weak | 45.0 | 41.8 | 38.3 | 34.0 | 29.7 | 26.3 | 22.1 | 15.7 | 4.7 | 1.0 | 25.9 |

## 4.5 COMPARISON WITH THE STATE-OF-THE-ART

**Audio-visual event localization:** In Table 5, we compare the proposed method with the recent fully and weakly-supervised methods on the AVE dataset for audio-visual event localization task. In the weakly-supervised setting, our method performs better than all of the existing methods at least by 1.4%. Note that, even though learned in weak-supervision, our approach achieves a comparable accuracy (77.1%) to the fully-supervised accuracy of the state-of-the-art method (Xuan et al., 2020).

**Temporal action localization:** In Table 6, we apply the proposed method to weakly-supervised action localization in long duration videos of the ActivityNet1.2 dataset. We report results for our method as well as its efficient version from Section 4.4. The mAP scores at varying IoU thresholds are compared with the current state-of-the-art methods. Both our method and its efficient version achieve the highest mAPs for 8 out of 10 IoU thresholds, and outperform all of the previous methods with the avg. mAP of 26.0%. We also significantly outperform the audio-visual based method of Tian et al. (2018) by the avg. mAP of 17.2%. Additionally, we compare with two naive fusions without the cross-attention (0-stage, SoftMax) with and without the continuity losses (denoted as CL in the Table), both are bettered comfortably by our method. This demonstrates that the effective fusion of audio and visual modalities is critical for action localization. Furthermore, our approach is even comparable to the fully-supervised method in (Zhao et al., 2017).

## 5 CONCLUSION

We presented a novel approach for weakly-supervised temporal action localization in videos. In contrast to other methods, we leveraged both audio and visual modalities for this task. This is the first attempt at audio-visual localization of unconstrained actions in long videos. To collaboratively fuse audio and visual features, we developed the multi-stage cross-attention mechanism that also preserves the characteristics specific to each modality. We proposed to use the open-max classifier to model the action foreground and background, in absence of temporal annotations. Our model learns to classify video snippets via two consistency losses that enforce continuity for foreground reliability and open-max probabilities for action classes and the background. We conducted extensive experiments to analyze each of the proposed components and demonstrate their importance. Our method outperforms the state-of-the-art results on both AVE and ActivityNet1.2 datasets.

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

## A   ANALYSIS ON MULTI-STAGE CROSS-ATTENTION

In this section, we conduct extensive analysis for the impact of the multiple stages and dense skip connection of the proposed cross-attention mechanism. Tables 7 and 8 show the experimental results.

**Training multiple stages of cross-attention:**    As shown in the Table 1, the 3-stage model suffers from performance drop. To analyze this, in Table 7, we compare 2- and 3-stage models on each of 'w/o skip connection', 'w/ skip connection', and 'w/ dense skip connection'. Without the skip connection, 3-stage model improves over 2-stage model, which is intuitively expected. With the skip connection, avg. mAP of 3-stage model drops compared to 2-stage model, from 24.9% to 23.2%. But, when the third stage is appended to the trained (and now frozen) stages of 2-stage model, the avg. mAP is maintained at 24.9%. Similarly, with the dense skip connection, training the entire 3-stage model end-to-end leads to degraded performance. But, when training the model frozen till the second stage the drop is much less. The fact that, in 3-stage model, better performance is obtained when training with first two stages frozen compared to training end-to-end, shows that the optimization gets hard in the latter. Therefore, we conclude that though the third stage helps without the skip connections, due to harder optimization with more stages and (dense) skip connections, 2-stage model is the optimal choice.

Table 7: **Comparing 2-stage and 3-stage models:** Training of 3-stage model is analyzed with skip connections in comparison to the 2-stage model.

| Method | Model | Avg. mAP |
|---|---|---|
| w/o skip connection | 2-stage | 24.1 |
|  | 3-stage | 24.7 |
| w/ skip connection | 2-stage | 24.9 |
|  | 3-stage | 23.2 |
|  | 3-stage (frozen till $2^{\text{nd}}$ stage) | 24.9 |
| w/ dense skip connections | 2-stage | 26.0 |
|  | 3-stage | 23.3 |
|  | 3-stage (frozen till $2^{\text{nd}}$ stage) | 25.2 |

Table 8: **Need for multi-stage:** Ablation study on the size of the dimension for the cross-correlation matrix $W \in \mathbb{R}^{d_{x,u} \times d_{x,v}}$ for 1-stage model on the AVE dataset. 2-stage model with $W \in \mathbb{R}^{1024 \times 1024}$ is the proposed.

| Method | $d_{x,u}$ | $d_{x,v}$ | Accuracy(%) |
|---|---|---|---|
|  | 128 | 512 | 72.7 |
|  | 128 | 1024 | 73.2 |
|  | 128 | 2048 | 73.7 |
|  | 512 | 512 | 74.8 |
|  | 512 | 1024 | 74.8 |
|  | 512 | 2048 | 74.7 |
| 1-stage | 1024 | 512 | 73.2 |
|  | 1024 | 1024 | 75.0 |
|  | 1024 | 2048 | 75.0 |
|  | 2048 | 512 | 73.7 |
|  | 2048 | 1024 | 74.3 |
|  | 2048 | 2048 | 74.4 |
| 2-stage | 1024 | 1024 | 77.1 |

**Need for multi-stage cross-attention:**    In Table 8, we experiment with 1-stage model, varying the size of dimensions ($d_{x,u}$ and $d_{x,v}$) of the cross-correlation matrix $W$ on the AVE dataset. We tried several hyper-parameter settings in 1-stage model, but none of them could outperform the default

setting ($d_{x,u} = 1024$, $d_{x,v} = 1024$) of 2-stage model even with more parameters. Instead of increasing the parameters in 1-stage model itself, when an additional stage is added (i.e. a weight matrix learned with a non-linear activation function) better performance is achieved. Indeed, it is often not trivial to replace a sequence of non-linear functions with another non-linear function as we experimentally observe here. The intention behind the multi-stage is also to extensively delve into cross-modal information, progressively learning the embeddings for each modality.

# B ANALYSIS FOR CONSISTENCY LOSSES AND OPEN-MAX CLASSIFICATION

## B.1 ANALYSIS OF CONSISTENCY LOSSES ON DIFFERENT STAGE MODELS

In Table 9, we conduct the analysis for the consistency losses for 0, 1 and 3-stage models as well as the chosen 2-stage model.

**Effect of losses on different stage models:** The impact of continuity losses is analogous on 1-, 2- and 3-stage models. Each of the two continuity losses help, but the pseudo localization loss ($\mathcal{L}_{\text{pseu}}$) is more effective. Also, there is further benefit of using them together for almost all the IoU thresholds and stages. In 0-stage model, i.e. without the cross-attention, O-II shows the highest snippet-level performance on the AVE dataset, but the lowest temporal action localization performance on the ActivityNet1.2 dataset. From this, we understand that $\mathcal{L}_{\text{pseu}}$ has difficulty in achieving continuity when audio and visual features are overly heterogeneous. Consequently, clear benefit is observed when the cross-attention is used.

**Interdependence of cross-attention and pseudo localization loss:** When comparing the O-I of all 0-3 stage models, we see that the performance improvement by stacking the cross-attention is marginal, and the pseudo localization is critical to the performance. This follows from Eq. 9, where $\mathcal{L}_{\text{pseu}}$ is only activated at snippet $l$ when classification over its neighboring snippets does not strongly agree on the action class or background. To analyze this, we check how frequently $\mathcal{L}_{\text{pseu}}$ is activated when cross-attention is not used and when it is used. For 0-stage model, without continuity losses, $\mathcal{L}_{\text{pseu}}$ is activated on 11.1% snippets of the ActivityNet1.2 training set. The same frequency is 38.2% for 2-stage model, again without the continuity losses. This shows that when the cross-attention is used, more often the open-max classification of a snippet fails to strongly agree with its neighbors. Therefore, the pseudo localization loss is much needed to enforce the continuity.

Table 9: For 0-3 stage models, ablation analysis on consistency losses for open-max (O-0, O-I, O-II, and O-III) classifiers on the AVE and the ActivityNet1.2 datasets. O-III of 2-stage is the proposed.

| | Method | $\mathcal{L}_{\text{cont}}$ | $\mathcal{L}_{\text{pseu}}$ | AVE | ActivityNet1.2 [mAP@IoU (%)] | | | | |
|---|---|---|---|---|---|---|---|---|---|
| | | | | Accuracy (%) | 0.5 | 0.6 | 0.7 | 0.8 | Avg. |
| 0-stage | O-I | ✓ | | 57.7 | 42.8 | 35.9 | 27.6 | 7.2 | 21.0 |
| | O-II | | ✓ | 66.8 | 29.4 | 25.3 | 20.5 | 13.6 | 16.5 |
| | O-III | ✓ | ✓ | 65.0 | 37.6 | 32.4 | 26.7 | 19.4 | 22.0 |
| 1-stage | O-I | ✓ | | 64.6 | 39.8 | 33.4 | 27.4 | 13.0 | 21.9 |
| | O-II | | ✓ | 73.7 | 41.5 | 34.9 | 28.4 | 20.9 | 24.0 |
| | O-III | ✓ | ✓ | 75.0 | 42.1 | 35.3 | 29.5 | 20.8 | 24.1 |
| 2-stage | O-I | ✓ | | 64.9 | 39.9 | 33.7 | 23.8 | 14.3 | 20.3 |
| | O-II | | ✓ | 75.9 | 44.1 | 37.4 | 31.1 | 22.4 | 25.7 |
| | O-III | ✓ | ✓ | 77.1 | 44.8 | 37.8 | 30.8 | 22.5 | 26.0 |
| 3-stage | O-I | ✓ | | 66.2 | 38.4 | 31.8 | 25.5 | 17.7 | 21.5 |
| | O-II | | ✓ | 74.3 | 39.8 | 33.7 | 27.9 | 20.7 | 23.2 |
| | O-III | ✓ | ✓ | 74.6 | 39.5 | 33.8 | 27.9 | 20.9 | 23.3 |

## B.2 ANALYSIS OF LOSSES AND OPEN-MAX CLASSIFICATION ON 2-STAGE MODEL

In Table 10, we conduct more extensive analysis for the consistency losses and the open-max classifier. Specifically, we replace the open-max classification approach with soft-max one. Then, for both classifiers with the 2-stage cross-attention, we ablate the foreground continuity or pseudo local-

ization losses where CAS and MIL losses are commonly used. First, the performance gap between S-0 and O-0, where only CAS and MIL losses are used, shows the difficulty of learning two parallel branches in weakly-supervised manner. However, when adding the pseudo localization loss, (S-I and O-II), the open-max classification approach is further improved than the soft-max. Hence, the pseudo labels reduce the fallacious action classification of snippets and are more effective on the open-set background modeling than the closed-set modeling.

Next, O-I and O-II shows higher performance than O-0. Similarly, S-I is superior to S-0. This indicates that erroneous classifications are suppressed by the correctly classified neighbors when using the consistency losses. Also, comparing O-I and O-II, the pseudo localization loss gives more performance improvement. This is because the pseudo localization loss addresses the consistency of classification scores of all the classes including background, while the foreground continuity loss smoothens foreground reliability being class-agnostic or only for the target class. For all of the IoU thresholds (except 0.7), O-III, open-max classification with both of the consistency losses, yields the highest performance. Therefore, all of the proposed open-max classification and consistency losses are effective to temporal action or event localization in videos.

Table 10: Ablation analysis of 2-stage model on consistency losses for soft-max (S-0 and S-I) and open-max (O-0, O-I, O-II, and O-III) classifiers on the AVE and the ActivityNet1.2 datasets. O-III of is the proposed.

| Method | $\mathcal{L}_{\mathrm{cont}}$ | $\mathcal{L}_{\mathrm{pseu}}$ | AVE Accuracy (%) | ActivityNet1.2 [mAP@IoU (%)] | | | | |
|---|---|---|---|---|---|---|---|---|
| | | | | 0.5 | 0.6 | 0.7 | 0.8 | Avg. |
| S-0 | | | 62.1 | 36.4 | 28.4 | 22.7 | 15.8 | 19.6 |
| S-I | | ✓ | 68.5 | 39.4 | 35.7 | 30.7 | 19.8 | 23.8 |
| O-0 | | | 60.4 | 35.4 | 27.5 | 22.9 | 12.7 | 18.7 |
| O-I | ✓ | | 64.9 | 39.9 | 33.7 | 23.8 | 14.3 | 20.3 |
| O-II | | ✓ | 75.9 | 44.1 | 37.4 | 31.1 | 22.4 | 25.7 |
| O-III | ✓ | ✓ | 77.1 | 44.8 | 37.8 | 30.8 | 22.5 | 26.0 |

## C  DETAILS OF THE PROPOSED ARCHITECTURE

| Layer/Operation | No. parameters | Input | Output |
|---|---|---|---|
| $f_u$ + LeakyRelu | $d_u \times d_x$ | $U$ | $X_u \in \mathbb{R}^{d_x \times L}$ |
| $f_v$ + LeakyRelu | $d_v \times d_x$ | $V$ | $X_v \in \mathbb{R}^{d_x \times L}$ |
| $W^{(1)}$ | $d_x \times d_x$ | $X_u, X_v$ | $\tilde{X}_u^{(1)} \in \mathbb{R}^{d_x \times L}, \tilde{X}_v^{(1)} \in \mathbb{R}^{d_x \times L}$ |
| Dense Connection + Tanh | - | $X_u, X_v, \tilde{X}_u^{(1)}, \tilde{X}_v^{(1)}$ | $X_{\mathrm{att},u}^{(1)} \in \mathbb{R}^{d_x \times L}, X_{\mathrm{att},v}^{(1)} \in \mathbb{R}^{d_x \times L}$ |
| $W^{(2)}$ | $d_x \times d_x$ | $X_{\mathrm{att},u}^{(1)}, X_{\mathrm{att},v}^{(1)}$ | $\tilde{X}_u^{(2)} \in \mathbb{R}^{d_x \times L}, \tilde{X}_v^{(2)} \in \mathbb{R}^{d_x \times L}$ |
| Dense Connection + Tanh | - | $X_u, X_v, X_{\mathrm{att},u}^{(1)}, X_{\mathrm{att},v}^{(1)}, \tilde{X}_u^{(2)}, \tilde{X}_v^{(2)}$ | $X_{\mathrm{att},u}^{(2)} \in \mathbb{R}^{d_x \times L}, X_{\mathrm{att},v}^{(2)} \in \mathbb{R}^{d_x \times L}$ |
| Concatenation | - | $X_{\mathrm{att},u}^{(2)}, X_{\mathrm{att},v}^{(2)}$ | $X_{\mathrm{att}} \in \mathbb{R}^{2d_x \times L}$ |
| $f_{\mathrm{ac}}$ + Soft-Max | $2d_x \times C$ | $X_{\mathrm{att}}$ | $(p_{\mathrm{ac}}^l)_{l=1}^L \in \mathbb{R}^{C \times L}$ |
| $f_\mu$ + Sigmoid | $2d_x \times 1$ | $X_{\mathrm{att}}$ | $(\mu^l)_{l=1}^L \in \mathbb{R}^{1 \times L}$ |

## D  MULTIPLE INSTANCE LOSS AND CO-ACTIVITY SIMILARITY LOSS

We apply multiple-instance learning loss for classification. The prediction score corresponding to a class is computed as the average of its top $k$ activations over the temporal dimension. Co-activity similarity loss (CASL) (Paul et al., 2018) is computed over two snippet sequences from a pair of videos, to have higher similarity when the videos have a common class.

## E  QUALITATIVE EVALUATION

We provide additional qualitative results for action localization on the ActivityNet1.2 dataset. Fig. 4 compares the proposed method with the method trained on visual modality ('Visual-only'). The

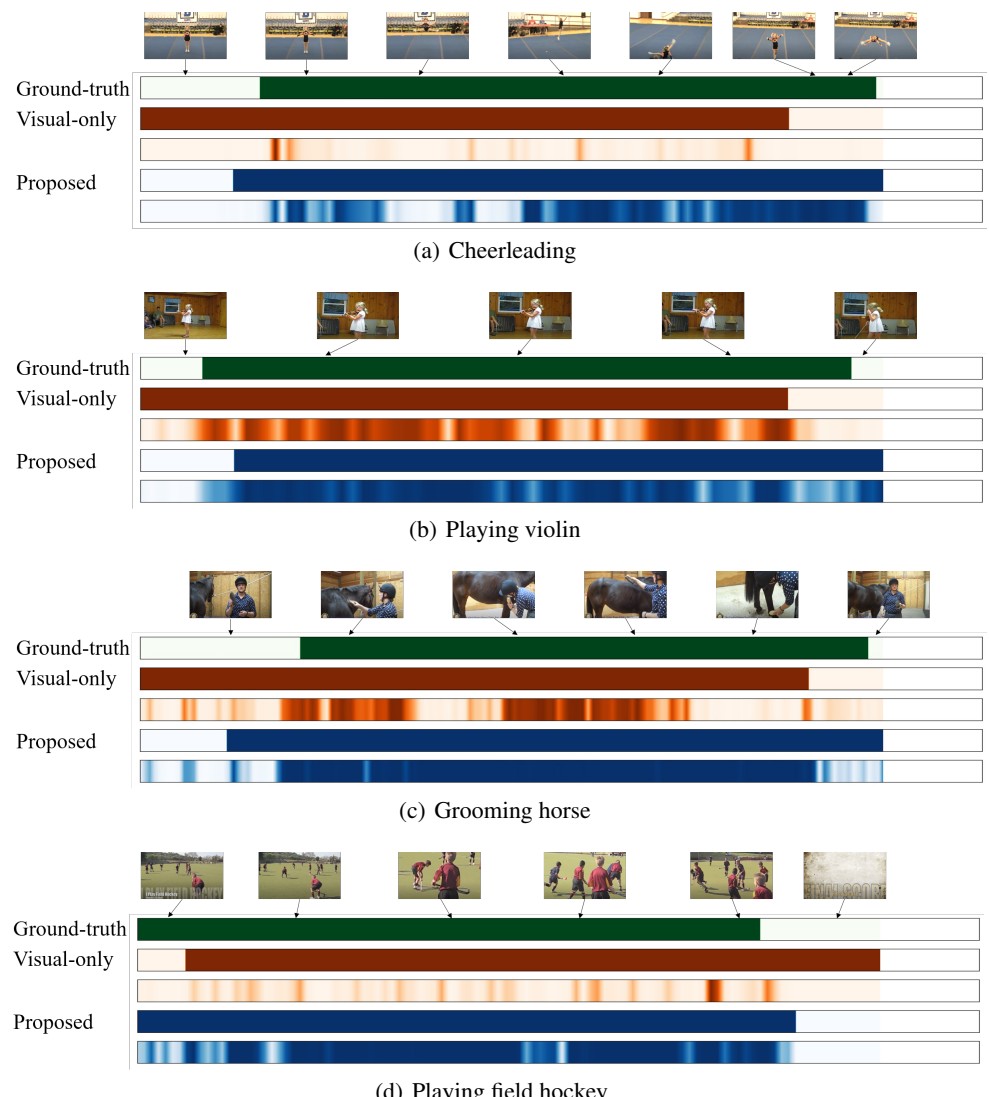

Figure 4: Qualitative results for action localization. Ground-truth (green), prediction by the visual-only method (orange), and prediction by the proposed method (blue) are shown. Class activation sequences are visualized below each prediction, darker shade means higher activation.

open-max classifier and total loss function are commonly used for both. In Figs. 4(a) and (b), because the videos are static in visual modality, the background segments in early parts of videos are miss-localized as actions in the visual-only model. Contrarily, proposed method distinguishes the background based on the action-related audio (cheerleading music and violin sound). In Fig. 4(c), the brushing sound is overlapped with the loud human narration lasting throughout videos. Nevertheless, the proposed method effectively extracts the crucial audio cues and fuses them with the visual ones. In Fig. 4(d), even though the early part of the action is visually occluded by large logos, our method exactly localizes the action. Also, for all of the class activation sequences, the activations by the proposed method are more consistently high for actions. This means that our collaboration of audio and visual modalities is more robust in distinguishing foreground from background.

Fig. 5 illustrates the cases where audio degrades the performance. Fig. 5 (a) shows an example video for action class 'playing violin'. The violin sound of the soloist and the band is intermingled in the video. In the end, the sound of violin continues making our model predict the action but since camera focuses on the band, the ground-truth does not include those frames. Fig. 5 (b) shows an example of action 'using parallel bars'. Here the repeated background music is irrelevant to action,

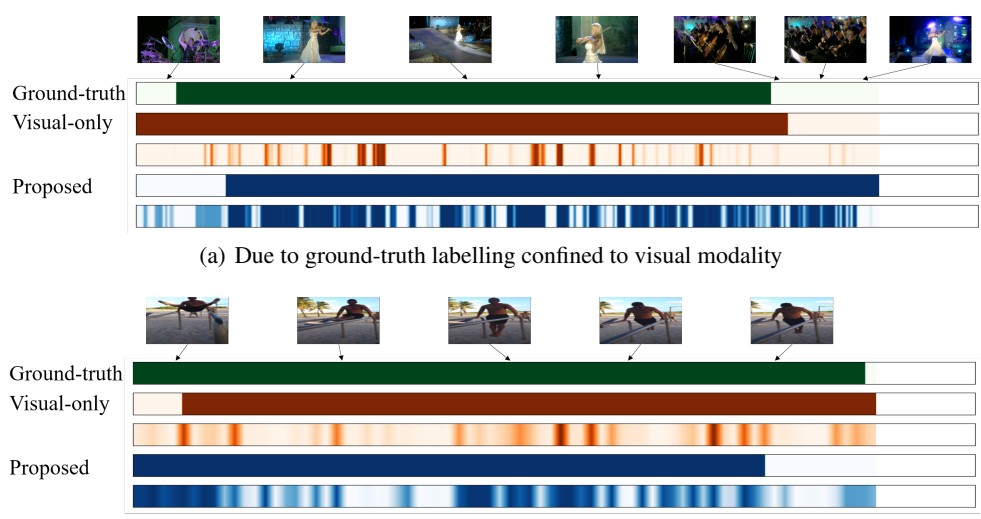

(a) Due to ground-truth labelling confined to visual modality

(b) Due to background music repeated for a long time

Figure 5: Examples where localization performance is degraded by audio.

therefore the class activation is bit off in the last part. However, thanks to the visual modality, the prediction is still reasonable.

