# OpenReview forum: "Cross-Attentional Audio-Visual Fusion for Weakly-Supervised Action Localization"
_ICLR.cc/2021/Conference — ICLR 2021 Poster_

### Official Review · AnonReviewer3 · 2020-10-21
**The motivation of the skip connection between the cross attention modules is unclear.**

**Rating:** 7
**Confidence:** 5

**Review:**

The authors propose to learn richer audio-visual representations for weakly-supervised action localization. First, the authors propose a multi-stage cross-attention mechanism to collaboratively fuse audio and visual features. Extensive experiments on two publicly available video-datasets (AVE and ActivityNet1.2) show that the proposed method appears to be effective.
The authors did not give the explanation of the built network in more details. Moreover, the motivation of the skip connection between the cross attention modules is unclear.

---

> ### Author Response · Authors · 2020-11-23
> **To AnnoReviewer3 [Detailed description for the proposed network, Effect of dense skip connection]**
>
> Thanks for appreciating our method and experiments, and for providing constructive suggestions that helped us improve our paper.
>
> 1.	**Detailed description for the proposed network**: We have added a detailed and complete description of our network architecture. Please see the Table of Appendix-C.
> 2.	**Effect of dense skip connection**: The dense skip connection is useful to preserve modality-specific information. Applying cross-attention (please see Eq. 1 and 2) brings the audio and visual embeddings closer, while the dense skip connections enforces modality specific information (please see Eq. 3), more so with the dense skip connections. Using the cross-attention and the dense skip connection alternatively, the learning can find the right amount of compatibility between the two embeddings while preserving the modality specific information, in order to optimize for the classification and the continuity losses (consequently localization performance). A description is added at the end of Section 3.1 to motivate dense skip connections. Also, we experimentally validate this in the added Table 3 (Section 4.3), where dense skip connections improve avg. mAP from 24.1% to 26%.

---

### Official Review · AnonReviewer1 · 2020-10-26
**The authors propose solving the task of action localization by using a multi-stage cross attention to build a shared representation of audio and visual embeddings followed by an open-max classifier to properly handle no-action cases. They perform an extensive comparison with other recent state-of-the-art models and an ablation study, proving the validity of the proposed method.**

**Rating:** 6
**Confidence:** 5

**Review:**

The authors propose to mix the independent embeddings of audio and visual data by a set of cross-attention layers to the task of audio-visual event localization. They weigh the features based on the correlation between the representations of both modalities to obtain the output representation. After several layers like these, combined with dense skip connections, the final output is a multimodal representation of the input where the features have been individually learned for each modality based on information from the other one. The final features are concatenated and passed through an open-max classifier. Finally, they design different losses to help enforce coherence and continuity among the proposed labels throughout the video.

In this respect, the only thing that is not clear enough to me is the novelty of the proposed cross-attention layer. While there are some mentions to previous works of multi-modal representation learning, there is no explicit assessment of the contributions of the proposed cross-attention mechanism with respect to previous attempts to build similar representations, or even previous uses of cross-attention for other tasks.

Regarding the experiments, on one hand they perform ablation studies that confirm the contribution of some of the proposed losses and the cross-attention mechanism for building the audio-visual representations, as well as the use of an open-max classifier instead of a soft-max. On the other hand, while an extensive experimental comparison with other recent state-of-the-art models has been performed, proving that the model can improve previous results, it is not clear to me how much of that improvement is due to the use of specific backbones. For instance, the authors use a ResNet152 to extract the visual features, which is a fairly big network. It is unclear to me how much of the improvement over previous state-of-the-art is due to the use of specific backbone networks. In other words, given that even the baseline without cross-attention layers performs close to second, how much is the number of parameters and specific architecture design in charge of building those initial embeddings actually responsible for the final results?

Pros of the paper:
Extensive experimental results.
Improves state of the art.
Clearly written.
Cons of the paper:
Novelty of some contributions unclear.

---

> ### Author Response · Authors · 2020-11-23
> **To AnnoReviewer1 [Novelty of cross-attention, Choice of backbone]**
>
> Thanks for appreciating our method and experiments, and for providing constructive comments that helped us improve our paper.
>
> 1.	**Novelty of cross-attention**:
> - **Previous uses of cross-attention**: We added more related papers in the multi-modal representation learning part (3rd paragraph) of Section 2, to compare our cross-attention mechanism with the previous uses of cross-attention for other tasks. The three added references use cross-attention for few-shot learning (Hou et. al. 2019) and image-text matching (Lei et al 2018, Wei et al 2020). The most similar to our cross-attention mechanism is the cross-attention module of Hou et. al. 2019, which computes cross-correlation spatially between features maps of two images. Whereas our cross-attention is designed for video and is computed between two temporal sequences of different modalities.
> - **Assessment of the contributions of the proposed cross-attention**: Since the previous cross-attention methods are not designed to correlate the temporal sequences, we could not experimentally compare ours with them. However, we do compare with Tian et al. (in Tables 5 and 6) and Xuan et al. (in Table 5), both of them employ one-way audio-guided visual attention. Better performance of our method indicates the contribution of our cross-attention. We further experimentally validate the benefit of our cross-attention in Tables 1 and 6 in the main paper, and Table 9 in the Appendix. We revised 2nd paragraph of Section 2 to describe how our cross-attention differs from the only relevant two-way attention method of Wu et. al.: “Assuming single event videos, they detect the event-related snippet by matching the video-level feature of one modality with the snippet-level feature sequence of the other modality. Contrastingly, our cross-attention is over the temporal sequences from both the modalities and does not assume single-action videos”. They do not report weakly supervised results on AVE dataset, so experimental comparison couldn’t be done, also the single video assumption makes their approach inapt for ActivityNet1.2 dataset.
> - **The novelty of our cross-attention layer**, therefore, comes from: (a) the first such method that computes cross-attention over temporal sequences of different modalities, and (b) the first audio-visual fusion method for action localization in long videos. Please see the revised 2nd and 3rd paragraphs of Section 2, and Tables 1, 5, 6 and 9 for more details and comparisons.
>
> 2.	**Choice of backbone**:
> - **Fair comparison**: We choose the backbone for fair comparison with the competing methods. In Table 5, for AVE dataset, all the competing methods use ResNet-152 as backbone. In Table 6, for ActivityNet1.2, we used I3D model for extracting visual features as majority of competing methods do the same. We used the same audio feature extractor as used for AVE dataset. So, improvement over SOTA is due to the proposed method.
> - **How much initial embeddings are responsible for the results**: In Table 5, the ‘naïve fusion’ method also uses the proposed continuity losses to achieve avg. mAP of 22.9%, still relatively about 12% lower than our final avg. mAP of 26%. We added results for another naïve fusion which does not use the proposed losses, avg. mAP drops to 19.8%.

---

### Official Review · AnonReviewer4 · 2020-10-28
**Weakly labelled multi modal action localization in video**

**Rating:** 6
**Confidence:** 2

**Review:**

This paper introduces a method for predicting the class of an action in a video and localizing the range.
The focus is on using both video and audio modalities.
They use a cross modal "multi stage" attention mechanism, a background classifier, and losses designed to encourage temporal consistency in the output.

Reason for Score:

I would accept this paper.
The model shows good performance for weakly labelled, multi-model action recognition and localization.
The method is well described and would have immediate benefit in practical applications.

Pros:

Weakly supervised - avoiding the cost of detailed labelling is very important.
Multi Modal - Table 1 shows great advantage of using both modalities, and audio is nearly always preset in video. Table 2 shows ablations for the different parts of the loss.
The attention mechanism in the model is intuitive. (except the need for multiple stages - see cons).

Cons:

I is not clear multi-stage attention is necessary - one stage seems to do well. Did the authors try different hyperparameters (eg. units/dimensions of single attention block) for one-stage of attention. There may be enough room to get the extra 2 percent, and why we should need 2 stages of attention is not clear, what is the intended effect?

The related Work section is rather plain to read, as the differences with the current work are not explained. It is a long list, and interrupts the flow of the paper without showing how it relates. Also the first sentence in "Related Work" does not read well.

Second paragraph of introduction states that the work intends to "localize a wider range of actions". The AVE dataset does not do this - is this a reference to the ActivityNet1.2 dataset - does it contain noisier and "weaker" labels than AVE?

Section 3.2 "snippet-wisely" is understandable but strange. Also in Fig 2 text: "respectively without and with L_cont and L_pseu" is difficult to follow.

---

> ### Author Response · Authors · 2020-11-23
> **To AnnoReviewer4 [Need and intended effect of multi-stage cross-attention, Revision of related work]**
>
> Thanks for appreciating our method and evaluation, acknowledging its importance. We are also thankful for the constructive suggestions that helped us improve our paper.
> 1.	**Need and intended effect of multi-stage cross-attention**: As suggested, we tried several hyper-parameter settings in 1-stage model but could not outperform 2-stage model even with more parameters. This is shown in Table 8 in the Appendix-A, where we vary the dimensions of the cross-correlation matrix $W$ and report results for the AVE dataset. Instead of increasing the parameters in 1-stage model itself, when an additional stage is added (i.e. a weight matrix learned with a non-linear activation function) better performance is achieved. Indeed, it is often not trivial to replace a sequence of non-linear functions with another non-linear function as we experimentally observe here. We had only reported in Section 4.2 but now we also mention in the caption of Fig. 1 that non-linear activation is used in each stage. The intention is also to extensively delve into cross-modal information, progressively learning the embeddings for each modality.
> 2.	**Revision of related work**: We have modified the Section 2 in the paper according to the suggestion. We start with an introductory sentence and add how our work is different from the related works in each paragraph (mostly at the end). Also, a few more related papers are added for better comparison and completeness.
> 3.	**AVE vs ActivityNet1.2**: Yes, the reference is to the ActivityNet1.2 dataset, which contains “wider range of actions” compared to the AVE dataset. The difference is not in the labels but in the kind of videos and action classes. In AVE, the action classes are limited to those with strong audio cues in small videos (10-second), where each video includes only one action instance; here leveraging audio is easier. Whereas, in ActivityNet1.2, wider variety of action classes are present in longer videos (few minutes) with multiple action instances per video and often with weak audio cues, which makes action localization and leveraging audio harder. We revised Section 4.1 to better clarify this while describing ActivityNet1.2 dataset.
> 4.	**Sentence formation/language**: We fixed the text in Section 3.2 as “… are fed snippet-wise into the …” and modified the caption of Figure 2 as well.

---

### Official Review · AnonReviewer2 · 2020-10-29
**The authors propose a novel approach to audio-visual action localization.  They introduce a multi-stage cross attention approach to fuse audio-visual features and a consistency loss to enforce temporal continuity.  It is a good paper that could be improved with some additional experiments/analysis and improved clarity.**

**Rating:** 6
**Confidence:** 3

**Review:**


Summary

The authors propose a novel approach to audio-visual action localization.  They introduce a multi-stage cross attention approach to fuse audio-visual features and a consistency loss to enforce temporal continuity.

Strengths

The approach is well motivated and the technical contribution is clear.  The paper is well written and the contributions are evaluated against a number of recent SOTA approaches on two datasets for action localization.

Weaknesses/Concerns

I would like to see a discussion on the drop in performance in the 3-stage model compared to the 2-stage and 1-stage models in Table 1.  This drop is not intuitive so the authors should provide some analysis here.

In section 4.3 and Table 2, it seems O-I, O-II and O-III are used before being defined.  Are these results with the 2-stage model?  I would like to see the effect of using the different stage models with the different versions of the loss function.  This will help understand the different contributions to the final performance.

The model introduces increased computation and additional parameters.  What is the increase in computation?  This could be important for detection systems that prioritize a trade-off of performance and efficiency.

Recommendation and reasoning

This is a good paper that could be improved with some additional experiments/analysis and improved clarity.  I recommend acceptance.

---

> ### Author Response · Authors · 2020-11-23
> **To AnnoReviewer2 [Performance drop in 3-stage model, Effect of the continuity losses on different stage models,  Model efficiency]**
>
> Thanks for appreciating our motivation, technical contribution, and evaluation. We are also thankful for the constructive suggestions that helped us improve our paper.
>
> 1.	**Performance drop in 3-stage model**: Indeed, the performance drop with 3-stage model is counter intuitive. To analyze this, in Table 7 (Appendix-A), we compare 2- and 3-stage models on each of ‘w/o skip connection', ‘w/ skip connection', and ‘w/ dense skip connection'. Without the skip connection, 3-stage model improves over 2-stage model, which is intuitively expected. With the skip connection, avg. mAP of 3-stage model drops compared to 2-stage model, from 24.9% to 23.2%. But, when the third stage is appended to the trained (and now frozen) stages of 2-stage model, the avg. mAP is maintained at 24.9%. Similarly, with the dense skip connection, end-to-end training of 3-stage model leads to degraded performance. But, when training the model frozen till the second stage, the performance drop is much less. The fact that, in 3-stage model, better performance is obtained when training with first two stages frozen compared to training end-to-end, shows that the optimization gets hard in the latter. Therefore, we conclude that though the third stage helps without skip connections, due to harder optimization with more stages and (dense) skip connection, 2-stage model is the optimal choice here.
> 2.	**Clarity on Section 4.3 and Table 2**: Yes, in Table 2, the results are for 2-stage models. We make this clear in the caption of Table 2, and clearly define the O-I, O-II, and O-III in Section 4.3 before their use.
> 3.	**Effect of the continuity losses on different stage models**: We have conducted the ablation analysis on the consistency losses for 0-, 1-, 2-, and 3-stage models in Table 9 (Appendix-B). The impact of the continuity losses is analogous on 1-, 2- and 3-stage models. Each of the two continuity losses are useful, but the pseudo localization loss is more effective. Also, there is further benefit of using them together for almost all the IoU thresholds and stages. In 0-stage model, i.e. without the cross-attention, O-II shows the highest snippet-level performance on the AVE dataset, but the lowest temporal action localization performance on the ActivityNet1.2 dataset. From this, we understand that the pseudo localization loss has difficulty in achieving continuity when audio and visual features are overly heterogeneous. That is why, we observe clear benefit when cross-attention is used. Doing this suggested analysis, we also observed the interdependence of the cross-attention and the pseudo localization loss, and explain this in the Appendix-B.
> 4.	**Computation increase by adding audio (model efficiency)**: Indeed, as noted by the reviewer, the computation cost is increased by adding audio. We added a sub-section 4.4 on the model efficiency. We report the increase in computation and also suggest an efficient version of our method. This version maintains performance (25.9% vs 26%) while considerably reducing the computation cost from 4 MFLOPS to 1.7 MFLOPS, that is less than that of the default visual-only model (2.3 MFLOPS). Please see the added Table 4 for details. Thanks for the suggestion.

---

### Decision · Program_Chairs · 2021-01-07
**Final Decision**

**Decision:**

Accept (Poster)

**Comment:**

The paper focuses on the task of weakly supervised activity detection (WSAD). The proposed method combines various ideas together: **(i)** a cross-attention module for audio-visual information fusion and better representation, **(ii)** an open-max classifier to treat the background as an open set, and **(iii)** loss terms to encourage temporal continuity of action predictions. The experimental results on well-known benchmark datasets are promising as they beat out many other methods in the literature.

Based on the reviewers' comments, it is clear that the reviewers unanimously see value in the proposed methodology and the competitive results. To strengthen the paper, the authors are encouraged to provide a stronger validation of the contributions being claimed for the specific task being addressed. This would more concretely position the claimed contributions in the WSAD literature.